# Targeting Lysosomes to Reverse Hydroquinone-Induced Autophagy Defects and Oxidative Damage in Human Retinal Pigment Epithelial Cells

**DOI:** 10.3390/ijms22169042

**Published:** 2021-08-22

**Authors:** Samuel Abokyi, Sze-Wan Shan, Christie Hang-I Lam, Kirk Patrick Catral, Feng Pan, Henry Ho-Lung Chan, Chi-Ho To, Dennis Yan-Yin Tse

**Affiliations:** 1School of Optometry, The Hong Kong Polytechnic University, Hung Hom, Hong Kong, China; samuel.abokyi@connect.polyu.hk (S.A.); samantha.shan@polyu.edu.hk (S.-W.S.); christie.h.lam@connect.polyu.hk (C.H.-I.L.); kirk-patrick.catral@polyu.edu.hk (K.P.C.); feng.a.pan@polyu.edu.hk (F.P.); henryhl.chan@polyu.edu.hk (H.H.-L.C.); chi-ho.to@polyu.edu.hk (C.-H.T.); 2Department of Optometry & Vision Science, College of Health and Allied Science, University of Cape Coast, Cape Coast 00233, Ghana; 3Centre for Eye and Vision Research, 17W Hong Kong Science Park, Hong Kong, China

**Keywords:** age-related macular degeneration, hydroquinone, oxidative stress, autophagy, ubiquitin-proteasome system (UPS), lysosomal alkalization

## Abstract

In age-related macular degeneration (AMD), hydroquinone (HQ)-induced oxidative damage in retinal pigment epithelium (RPE) is believed to be an early event contributing to dysregulation of inflammatory cytokines and vascular endothelial growth factor (VEGF) homeostasis. However, the roles of antioxidant mechanisms, such as autophagy and the ubiquitin-proteasome system, in modulating HQ-induced oxidative damage in RPE is not well-understood. This study utilized an in-vitro AMD model involving the incubation of human RPE cells (ARPE-19) with HQ. In comparison to hydrogen peroxide (H_2_O_2_), HQ induced fewer reactive oxygen species (ROS) but more oxidative damage as characterized by protein carbonyl levels, mitochondrial dysfunction, and the loss of cell viability. HQ blocked the autophagy flux and increased proteasome activity, whereas H_2_O_2_ did the opposite. Moreover, the lysosomal membrane-stabilizing protein LAMP2 and cathepsin D levels declined with HQ exposure, suggesting loss of lysosomal membrane integrity and function. Accordingly, HQ induced lysosomal alkalization, thereby compromising the acidic pH needed for optimal lysosomal degradation. Pretreatment with MG132, a proteasome inhibitor and lysosomal stabilizer, upregulated LAMP2 and autophagy and prevented HQ-induced oxidative damage in wildtype RPE cells but not cells transfected with shRNA against ATG5. This study demonstrated that lysosomal dysfunction underlies autophagy defects and oxidative damage induced by HQ in human RPE cells and supports lysosomal stabilization with the proteasome inhibitor MG132 as a potential remedy for oxidative damage in RPE and AMD.

## 1. Introduction

Oxidative stress is a hallmark of several age-related disorders, including cardiovascular diseases, chronic obstructive pulmonary disease, chronic kidney disease, cancers, and neurodegenerative diseases [1]. Oxidative stress can lead to an accumulation of damaged and misfolded proteins and obsolete organelles, and eventually to dysregulation of cellular homeostasis and the development of human diseases [2,3,4]. More importantly, therefore, the quantification of reactive oxygen species (ROS) production alone is insufficient when evaluating oxidative stress because the status of the cellular antioxidative defense machinery also has to be considered [1,5].

Cellular catabolic pathways, including autophagy and the ubiquitin-proteasome system (UPS), play an antioxidative role as they regulate protein homeostasis, mitochondrial quality control, ROS production, and cellular adaptation [4,6]. Both pathways cooperate in maintaining proteostasis but are unique in substrate selection. The proteasome is efficient in degrading smaller and short-lived proteins, whereas larger and long-lived substrates are targeted for autophagy-lysosomal degradation [7]. Despite their unique roles in proteolysis, they are functionally connected [7,8]. The crosstalk and interplay between autophagy and the UPS have been described under conditions of cellular stress [7,8]. Evidence suggests that UPS inhibition can be compensated for by upregulating autophagy, but dysfunctional autophagy has been demonstrated to inhibit proteasome activity and dysregulate cellular homeostasis [7]. Moreover, upregulating proteasome activity was found to be ineffective as a protective mechanism against cellular stress mediated by impaired autophagy [7]. Therefore, depending on the changes in autophagy and proteasome activity by a compound, it may induce oxidative damage in cells or protect them against it.

The benzene metabolite hydroquinone (HQ) is an important environmental toxicant because of its widespread industrial application and health impact [9]. It has a high redox activity, resulting in increased ROS production and oxidative stress [9]. HQ exposure has also been identified as an underlying risk factor in cancerous diseases, inflammation, and neurodegenerative disorders [9]. Age-related macular degeneration (AMD), a central retinal disorder that causes visual impairment, is one of the neurodegenerative diseases for which the contribution of HQ has been well-studied. The aetiology of AMD is multifactorial, including genetic and environmental underpinnings, but it is commonly characterized by drusen and retinal pigment epithelial (RPE) abnormalities at the earliest pathophysiological stage [10]. Proteomic data from the retina of AMD donors’ eyes with drusen show highly enriched oxidative protein modifications, supporting oxidative stress as an important environmental risk factor for AMD [11]. Further, epidemiological data have shown that smoking cigarettes increases the risk for RPE oxidative damage and AMD [12] and increased plasma levels of HQ almost twice compared to non-smokers [13]. Under controlled laboratory conditions, chronic exposure to cigarette smoke or HQ in mice led to oxidative damage in RPE and AMD-like pathology [14,15,16]. Furthermore, in vitro HQ caused oxidative damage of cultured human RPE cells, leading to dysregulated vascular endothelial growth factor (VEGF) homeostasis and increased inflammatory cytokine levels, which mirror significant events in the pathogenesis of AMD [17,18].

On the basis that human RPE cells are highly resistant to multiple pro-oxidants [19] but comparatively prone to HQ damage, we hypothesized that changes in autophagy and/or proteasome activity may underlie HQ-toxicity. An improved understanding of HQ-induced effects on cellular antioxidative pathways may lead to therapeutic strategies for RPE protection against oxidative damage and the development of AMD. In the current study, HQ inhibited autophagy and induced proteasomal activity in human RPE cells, whereas the ROS inducer H_2_O_2_ had the opposite effect. Autophagy inhibition by HQ involved downregulation of the lysosomal membrane-stabilizing protein LAMP2, lysosomal enzyme cathepsin D, and lysosomal alkalization, indicating loss of lysosomal membrane integrity and function. Proteasome inhibition with MG132, however, stabilized lysosomes, induced autophagy, and ameliorated HQ-induced oxidative damage in wildtype RPE cells but not in shRNA ATG5 transfected cells. The current study demonstrates the involvement of autophagy-lysosomal dysfunction in HQ-induced oxidative stress, suggests crosstalk between autophagy and UPS, and highlights lysosomal stabilization, via proteasome inhibition, as an interventional strategy against RPE oxidative damage and AMD.

## 2. Results

### 2.1. Comparative Vulnerability of RPE Cells to HQ-Induced Oxidative Damage

HQ has been implicated in the oxidative damage of human RPE during AMD pathogenesis [14,16]. To confirm the vulnerability of human RPE cells to HQ-induced oxidative stress, assays for ROS levels, protein carbonyls, and cell viability were performed on cells incubated for 2 h with HQ or H_2_O_2_ [20]. A 2-h incubation interval was selected to reflect the elimination half-life of HQ in vivo, as HQ is rapidly metabolized, and the parent compound and metabolites are largely eliminated within 1 h [21]. HQ doses below 25 μM were used for investigating oxidative stress to avoid potential genotoxic effects of HQ that may occur at doses above 25 μM [22]. Moreover, this dose range corresponds with available data on HQ levels in the blood after benzene/HQ exposure [23]. Literature data indicate that the highest H_2_O_2_ level measured in human tissue or blood is about 35 µM [24]. However, we used up to 500 μM H_2_O_2_ because earlier reports and our preliminary studies suggested the relative resistance of human RPE cells to H_2_O_2_-induced oxidative stress [19]. Both HQ and H_2_O_2_ caused dose-dependent increases in ROS levels, protein carbonyl levels, and the loss of cell viability in RPE cells (Figure 1A,B). Interestingly, while HQ-induced elevation of ROS levels was notably much lower than with H_2_O_2_ (Figure 1A,B), HQ induced higher protein carbonyl levels and loss of viability (Figure 1B). For instance, 62.5 µM H_2_O_2_ generated a significant increase in ROS levels yet caused little or no effect on the protein carbonyl levels and cell viability. In contrast, 20 µM HQ generated a comparable increase in ROS levels and at the same time caused a significant increase in protein carbonyl levels (*p* < 0.001, one-way ANOVA) and loss of cell viability (*p* < 0.001; one-way ANOVA; Figure 1B). These results confirmed the relative vulnerability of RPE cells to HQ-induced stress.

### 2.2. HQ-Induced Mitochondrial Dysfunction

The mitochondrion is a primary source of intracellular ROS and, therefore, is affected early in oxidative stress-mediated cell death [25]. To demonstrate that oxidative stress was involved in HQ-induced RPE damage, we examined the impact of HQ on the mitochondrial morphology and mitochondrial membrane potential of RPE cells. Healthy mitochondria exist as a dynamic network of an interconnected tubular structure, and a compromise of the mitochondrial function distorts this arrangement, resulting in altered connectivity and formation of short, round mitochondria [26]. Moreover, the mitochondrial membrane potential is directly correlated with ATP production because it reflects the process of electron transport and oxidative phosphorylation [27]. Our results showed that cells incubated with HQ had marked disorganization of the mitochondrial network compared with control RPE cells (Figure 2A). Further, HQ exposure led to a significant reduction in the mitochondrial membrane potential of human RPE cells (Figure 2B), in agreement with recent studies [28]. While H_2_O_2_-treated cells also showed some changes in the mitochondrial network morphology (Figure 2A), the mitochondrial membrane potential remained unaffected after treatment with up to 500 µM H_2_O_2_ (Figure 2B), confirming the relative resistance of RPE cells against H_2_O_2_-induced damage. These data, therefore, underpin oxidative stress as the underlying cause of HQ-induced RPE damage.

### 2.3. HQ Impairs Autophagy Flux in RPE Cells

The autophagy-lysosomal pathway (ALP) participates in the cellular response to oxidative stress through the degradation of oxidized proteins, lipids, and damaged mitochondria, particularly in post-mitotic cells [4]. We investigated the role of the ALP in HQ- and H_2_O_2_ induced toxicity on RPE cells. Autophagosomes are double-membrane vesicles that sequester intracellular substrates for lysosomal degradation [29]. Their membrane contains lapidated, membrane-bound LC3 protein (LC3-II), a commonly used marker for autophagosomal staining. In transfected RPE cells expressing GFP-LC3, HQ and H_2_O_2_ increased the number of autophagosomes, as demonstrated by increased numbers of GFP-LC3 puncta, which reflect clusters of LC3-II in autophagosomes (Figure 3A). Furthermore, we observed that the endogenous LC3-II level increased dose-dependently when cells were treated with HQ (Figure 3B). At concentrations up to 250 µM, H_2_O_2_ also dose-dependently increased LC3-II, but at 500 µM, H_2_O_2_ caused LC3-II to decline (Figure 3C). [30]. Since the net formation and degradation of autophagosomes (or LC3-II) determine its expression level in a cell, an increase in autophagosome number (or LC3-II) may indicate autophagy upregulation or the block of autophagy flux [30,31].

To determine if altered protein levels of LC3-II after treatment with HQ and H2O2 were due to changes in the rate of autophagosome formation or lysosomal degradation the exact role of autophagy, we assessed the levels of LC3-II in the presence of the lysosomal inhibitor chloroquine (CQ), which blocks the later stage of autophagy flux, thereby facilitating selective quantification of treatment-induced autophagosome (or LC3-II) formation [32]. When the autophagy flux is blocked by CQ, an additional increase in LC3-II levels by a drug will indicate autophagosome induction. As expected, treatment of the RPE cells with 50 μM CQ for 8 h resulted in a dramatic increase in LC3-II levels compared to the control (Figure 3D) due to the inhibition of basal autophagy and LC3-II degradation [33]. Following CQ treatment, additional treatment with HQ did not affect LC3-II protein levels, indicating that HQ did not alter autophagosome formation but likely reduced autophagosome flux (Figure 3D). In stark contrast, co-incubation with H_2_O_2_ and CQ led to increased LC3-II levels relative to treatment with CQ alone (Figure 3E), indicating H_2_O_2_-induced upregulation of autophagosome formation. In RPE cells and other cell types, autophagy upregulation was found to be protective against H_2_O_2_-induced oxidative damage [34,35]. The H_2_O_2_-mediated induction of autophagosome formation and flux may explain comparatively lower protein carbonyl levels and mitochondrial dysfunction after H_2_O_2_ treatment, relative to HQ treatment, despite H_2_O_2_ leading to more ROS formation. Together, these results demonstrated that HQ, unlike H_2_O_2_, inhibited autophagy in human RPE cells, and relative to H_2_O_2_, CQ dysregulates cellular homeostasis more potently.

### 2.4. HQ Does Not Downregulate TFEB or Downstream Autophagy Genes

TFEB is the master transcription factor coordinating autophagy and lysosomal biogenesis through transcriptional regulation of a network of genes known as the CLEAR network [36]. Hence, to fully elucidate the role of HQ on the ALP, we sought to determine whether impaired TFEB activity was involved. TFEB activation leads to its nuclear translocation and binding to the promoter regions of the CLEAR network, resulting in TFEB overexpression and autophagy induction [37]. Expression levels of TFEB as well as its downstream target genes (including ATG5 and ATG7 involved in autophagosome formation) are good indicators of TFEB activity and, therefore, were evaluated. The data revealed that HQ elevated the protein expression of TFEB (Figure 4A) and the mRNA levels of ATG5 and ATG7 (Figure 4B). These data rule out HQ-mediated dysregulation of autophagy at the transcriptional level.

### 2.5. HQ Downregulates LAMP2 and Cathepsin D Expression

Autophagy inhibition resulting in autophagosome (or LC3-II) accumulation may arise from deficits in the autophagosome-lysosome fusion and/or lysosomal degradation [38]. Hence, we determined whether HQ impaired lysosomal function by monitoring changes in the lysosome membrane protein LAMP2, which is integral in the control of autolysosome formation and degradation of autophagosomes [38,39]. We found that HQ treatment led to a decline in LAMP2 levels in a dose-dependent manner (Figure 4C), suggesting that HQ impaired lysosomal function in RPE cells.

The lysosomal aspartic protease cathepsin D is also a relevant indicator of lysosomal activity as it is involved in proteolytic degradation of autophagy substrate [40]. LAMP2 downregulation affects cathepsin D since the former is involved in the trafficking of the latter to lysosomes via endosomes after synthesis by the rough endoplasmic reticulum and Golgi complex [39,40]. Moreover, LAMP2 deficiency also leads to the loss of lysosomal membrane integrity, resulting in reduced retention of cathepsin D within the lumen [41]. Given the intimate relationship between LAMP2 and cathepsin D, we determined the effect of HQ on cathepsin D protein expression in RPE cells. Consistent with reduced LAMP2 expression, the cathepsin D levels also dose-dependently decreased following HQ treatment (Figure 4D), suggesting that dysregulated LAMP2 expression also affected cathepsin D expression and lysosomal homeostasis.

### 2.6. HQ Induces Lysosomal Alkalization

Intra-lysosomal enzymes involved in substrate degradation function optimally within a narrow range of acidic pH values [42,43]. Therefore, lysosome-alkalizing substances significantly impair lysosomal function and autophagy. Alteration of the lysosomal pH affects both autophagy and the phagocytic RPE functions [42]. The effect of HQ on the intra-lysosomal pH was determined using a ratiometric probe that changed from blue to yellow fluorescence with increasing acidity. While the exposure of cells to 62.5 µM H_2_O_2_, about twice the physiological concentration of H_2_O_2_ found in the body, did not affect the lysosomal pH, HQ at concentrations as low as 2.5 µM significantly increased the luminal pH to 5.6 compared with the control 5.1 (Figure 5A, *p* < 0.001, one-way ANOVA, Dunnett’s post hoc test). Thus, exposure of RPE cells to HQ may cause lysosomal dysfunction through the loss of lysosomal membrane integrity (i.e., downregulation of LAMP2 and cathepsin D levels) and lysosomal alkalization.

### 2.7. HQ Upregulates Proteasome Activity in RPE Cells

Crosstalk between autophagy and the UPS is observed in many systems [7,8]. Inhibition of proteasome activity increases autophagy, supporting a compensatory regulation between these pathways under cellular stress conditions [44,45]. We therefore assessed the interplay between autophagy and UPS under oxidative stress using HQ and H_2_O_2_. We found that the two oxidants affected proteasome activity differently as HQ-treated cells showed increased proteasome activity (Figure 5B) whereas H_2_O_2_-treated cells had a decline in proteasome activity (Figure 5C). Our data confirm compensatory regulation between autophagy and UPS under oxidative stress in RPE cells with H_2_O_2_ causing proteasome and increased autophagy, and HQ treatment causing the opposite effect, consistent with literature findings [44]. However, these results challenge an earlier hypothesis that proteasome inactivation was a crucial event in the oxidative damage of RPE cells [46,47]. Our data demonstrated that compensatory proteasome activation in autophagy-deficient HQ-treated cells did not provide effective protection against oxidative stress.

### 2.8. MG132 Stabilizes Lysosomes, Improves Autophagy, and Protects against Oxidative Damage

Recently, the peptide aldehyde proteasome inhibitor MG132 was found to stabilize the lysosomal membrane, restore lysosomal pH homeostasis, and induce autophagy in macrophages [48]. In addition, MG132 was shown to enhance cathepsin D activity and elevate LAMP1 levels [45]. Therefore, we tested MG132 in HQ-treated RPE cells. Firstly, we performed a dose-response study to investigate the effect of MG132 on proteasome activity, ROS generation, and apoptosis in ARPE-19 cells. We observed that MG132 inhibited proteasome activity dose-dependently from 2.5 µM to10 µM (Figure 6A) without affecting ROS levels (data not shown), compared with the solvent control (i.e., ethanol). However, we found that treating cells with 10 µM MG132 induced apoptosis in RPE cells (Figure 6B).

Next, we assessed the effect of MG132 on autophagy in human RPE cells. Our results demonstrated that MG132 treatment increased LC3-II in a dose-dependent manner including in the presence of CQ (Figure 7A,B), consistent with MG132 increasing autophagosome formation. Moreover, MG132 upregulated LAMP2, a lysosomal membrane protein involved in lysosomal stabilization and homeostasis (Figure 7C). Thus, our results corroborated the efficacy of MG132 in the stabilization of lysosomes and induction of autophagy [45,48].

Pre-treatment of RPE cells with MG132 ameliorated the toxic effects of 25 µM HQ. Specifically, HQ-induced apoptosis, protein carbonylation, and mitochondrial depolarization were significantly reduced with MG132 pre-treatment (Figure 8A), supporting its protection against HQ-induced oxidative damage. The lowest dose of MG132 (i.e., 2.5 µM) most effectively reduced HQ-induced apoptosis likely because MG132, at higher doses, had a pro-apoptotic effect itself.

To confirm that the cytoprotective effects of MG132 were mediated through increased autophagosomal flux, autophagy-defective cells expressing shRNA against ATG5 (Appendix A) were used for the additional experiment. In autophagy-deficient cells, the cytoprotective effect of MG132 against HQ-toxicity was completely lost with a higher number of the HQ- and M132-treated cells progressing from early to late apoptosis relative to cells treated only with HQ (Figure 8C). Thus, in the absence of compensatory upregulation of the ALP, proteasome inhibition with MG132 in itself does not confer any protection against HQ-induced toxicity [8].

## 3. Discussion

Oxidative damage induced by hydroquinone affects different cells in the body, contributing to several human diseases. The RPE is a specialized epithelium interfaced between the neuroretina and choriocapillaris. This layer is critical in the maintenance of retinal homeostasis due to its multifunctional roles, which include the transport of nutrient and metabolic waste between the neuroretina and choriocapillaris, the re-isomerization of visual pigments involved in phototransduction, phagocytosing of shed photoreceptor outer segments, and its protection against photooxidation [49,50]. The present study confirmed earlier work on the RPE’s relative resistance to oxidative stress from H_2_O_2_ and relative vulnerability to HQ [14,15]. We further investigated the mechanisms involved in hydroquinone-induced RPE oxidative damage because insight into such mechanisms may allow the formulation of therapeutic strategies for RPE protection and potential for the treatment of AMD. This study highlights the central role of the ALP in the response of RPE to HQ-induced oxidative stress. We demonstrated that HQ impaired lysosomal function and autophagy via lysosomal alkalization and disruption of lysosomal membrane integrity. These events led to the accumulation of damaged proteins, mitochondrial dysfunction, and RPE cell apoptosis. Our results also provided evidence that safeguarding lysosomes was an effective interventional approach to preventing HQ-induced autophagy dysfunction and oxidative damage in RPE cells.

### 3.1. Autophagy and Its Protective Role in Oxidative Stress

Autophagy, in general, can act either as a pro-life or pro-death mechanism; hence, its role in AMD development could be inhibitory or progressive [51]. The antioxidative role of autophagy, however, is one of the pathways contributing to the promotion of cell survival under stressful conditions. Emerging evidence demonstrates the protective role of autophagy in several oxidative stress-linked neurodegenerative disorders, including Alzheimer’s disease, Parkinson’s disease, Huntington’s disease, and motor neuron diseases [52]. Oxidative stress causes protein damage and undermines mitochondrial quality control mechanisms, while autophagy exerts protective effects by eliminating toxic protein aggregates and defective mitochondria, thereby restoring cell and tissue homeostasis [53]. In this study, HQ-induced autophagy deficits in RPE cells, leading to autophagosome accumulation, protein oxidation, mitochondrial dysfunction, and apoptosis. The findings of the present in vitro study mirror the post-mortem findings in human donors with AMD, which include toxic protein aggregation, mitochondria defects, and poorly degraded autophagic substrates in the RPE [53]. In addition, chronic HQ exposure of malignant tumors and cancer cell lines of human origin demonstrate a central role of autophagy in promoting cell resistance against chemotherapy [54,55,56]. Autophagy inhibition usually led to the loss of their resistance against anticancer treatments, leading to apoptosis in HQ-induced cancer cells [54,55,56]. However, upregulation of autophagy and cytoprotection in cancer cells hints at a different role of HQ in the development of cancers relative to AMD development. Due to the similarities of cellular pathology observed in HQ-treated RPE cell culture and AMD, we believe that the findings of lysosomal dysfunction in this study are also relevant to the understanding of AMD pathogenesis. Hence, interventions that improve lysosomal function and mitigate autophagy inhibition in RPE cells may also hold potential as therapeutic strategies in AMD management.

### 3.2. Lysosomal Dysfunction Underlying HQ-Induced Autophagy Deficit and RPE Damage

AMD donor eyes show evidence of dysfunctional autophagy and decline in lysosomal activity, believed to be crucial in drusen formation and the pathogenesis of the disease [57]. Our results pointed to lysosomal dysfunction as the cause of the autophagy deficit in HQ-induced oxidative stress. This is consistent with recent emphases on the importance of lysosomal homeostasis in promoting autophagy and cell survival under oxidative stress [58,59]. Lysosomal degradation of a substrate in RPE cells is dependent on lysosomal hydrolases with acidic pH optima between 4.5 and 5.2 [42,43]. A low luminal pH requires an intact lysosomal membrane barrier composed of ubiquitous highly glycosylated, lysosome-associated membrane proteins including LAMP1 and LAMP2 [60]. LAMP2 also regulates the intracellular transport of hydrolases to lysosomes after sorting from the trans-Golgi network, and lysosomal membrane fusion with autophagosomes [60]. While dysregulated LAMP1 expression does not cause apparent lysosomal defects, LAMP2 deficiency impairs lysosomal degradation and autophagy and is implicated in Danon disease [60]. Therefore, based on the HQ-mediated decline in LAMP2 and cathepsin D levels and the alkalizing effect of HQ on lysosomes, the results of this study supported that loss of lysosomal membrane integrity and lysosomal function were critical events in mediating the toxicity of HQ in RPE cells [57,58,60]. Similar harmful effects of HQ on lysosomes as observed with HQ have also been observed with prooxidants including paraquat and lipofuscin, substances that inhibit autophagy and phagocytosis [58,59].

We ruled out the possibility of transcriptional downregulation of autophagy by demonstrating that HQ induced overexpression of TFEB and the activation of autophagy genes. The overexpression of TFEB in cells exposed to HQ may be a compensatory mechanism to induce lysosomal biogenesis [61]. However, TFEB activation may not reverse the autophagy deficit because it primarily controls the transcriptional activation of the ALP, whereas HQ may also affect the ALP at the protein level, in addition to its downstream lysosomal alkalizing effect.

### 3.3. Role of Autophagy and UPS Crosstalk in HQ-Induced Oxidative Stress

Under oxidative stress in RPE cells, previous reports indicated that the UPS is compromised, implicating the downregulation of UPS in RPE damage [47]. Our data, on the contrary, showed increased proteasome activity when human RPE cells were exposed to HQ. It is known, however, that the effect of an oxidant on the UPS depends on the severity of ROS levels and whether the exposure is transient/sustained [6]. Hence, the differential effects between HQ and H_2_O_2_ on proteasome activity may be related to differences in their ROS generating capacities, which was much lower with HQ treatment for the tested concentrations. Furthermore, we demonstrated that proteasome inhibition with MG132 stabilized lysosomes, induced autophagy, and protected RPE cells from oxidative damage. Thus, the evidence in this study suggested compensatory crosstalk between the UPS and autophagy and supported the potential benefit of proteasome inhibition with MG132 in the management of AMD.

### 3.4. Conclusions

Overall, this study identified lysosomal dysfunction and autophagy deficits as the mechanisms underlying HQ-induced oxidative damage in human RPE. From our data, therefore, we propose treatments targeted at promoting lysosomal homeostasis and autophagy via proteasome inhibition, as potential therapeutic strategies in the management of AMD. While this strategy is feasible in our experiments, further studies to ascertain the criteria and the extent to which the cross-talk between autophagy and proteasome activity could be adequately altered without any harmful effects are warranted since increased proteasome inhibition or overstimulated autophagy with higher doses of MG132 offered no cytoprotection.

## 4. Materials and Methods

### 4.1. Cell Culture and Treatment

Human RPE cells (ARPE-19 cell line, ATCC^®^ CRL2302™) were cultured with Dulbecco’s modified Eagle’s medium (DMEM)/F12 (Sigma-Aldrich, St. Louis, MO, USA) containing 10% fetal bovine serum (Invitrogen-Gibco, Grand Island, NY, USA) and 1% penicillin/streptomycin antibiotic mixture (Thermal Fisher Scientific, Rockford, IL, USA). The cell line was thoroughly tested for mycoplasma using three different methods—agar culture (direct) method, Hoechst DNA stain (indirect) method, and PCR assay (lot #: 70022669, ATCC^®^). The medium was renewed every three days, and the cells were incubated at 37 °C in a humidified atmosphere containing 5% CO_2_.

For experiments, cells grown to 80% confluency between passage 11 and 16 were used as recommended by the supplier. After 24 h of serum starvation, cells were incubated with HQ (hydroquinone, H9003, Sigma-Aldrich), H_2_O_2_ (hydrogen peroxide 30%, 107209, Merck Millipore, Burlington, MA, USA), CQ (chloroquine diphosphate salt, C6628, Sigma-Aldrich), MG132 (M8699, Sigma-Aldrich), or a combination of these treatments. The whole-cell lysate was prepared using ice-cold 1 × RIPA lysis buffer [0.5 M Tris-HCl (pH 7.4), 1.5 M NaCl, 2.5% deoxycholic acid, 10% NP-40, and 10 mM EDTA (Millipore)] containing 1:100 protease inhibitor cocktail (Thermo Scientific, Waltham, MA, USA) unless otherwise stated. Bio-Rad Protein Assay was used to quantify sample protein concentrations (Bio-Rad Laboratories).

### 4.2. Cell Viability Assay

The Trypan blue dye exclusion assay was used to assess cell viability. Briefly, ARPE-19 cells with a seeding density of 1 × 10^6^ cells/well cultured in 6-well plates were incubated with treatments as desired in triplicate. After trypsinization and centrifugation at 1500 rpm for 5 min, cells were stained with 0.4% trypan blue solution (Sigma-Aldrich, T6146) to quantify viability as a percentage of the control.

### 4.3. Intracellular ROS Assay

Cells were plated on 96-well plates (1.5 × 10^4^ cells/well) or 35-mm MatTek glass-bottom dishes (MatTek Corp., Ashland, MA, USA) (1 × 10^6^ cells/dish) overnight. On the next day, cells were rinsed with PBS and incubated with 5 μM 5-(and-6)-chloromethyl-2′,7′-dichlorodihydrofluorescein diacetate (CM-H2DCFDA, C6827, Invitrogen, NY, USA) for 1 h in the dark at 37 °C. Afterward, cells were treated with HQ or H_2_O_2_ for 2 h or MG132 for 5 h. Intracellular ROS accumulation leads to increased fluorescence of cells due to the conversion of the cell-permeable dye from the non-fluorescent dichlorodihydrofluorescein (DCFH) to a highly fluorescent dichlorofluorescein (DCF) form within cells. A Clariostar microplate reader (BMG Labtech, Offenburg, Germany) or confocal microscopy (Eclipse Ti2-E, Nikon Instruments Europe B.V., Amsterdam, The Netherlands) was used to assess the fluorescence intensity at excitation/emission wavelengths of 483/530 nm. Data were normalized with Hoechst stain.

### 4.4. Live Cell Intra-Lysosomal pH Measurement

The ratiometric fluorescent dye LysoSensor™ Yellow/Blue DND-160 (cat # L7545, Thermo Fisher Scientific) was used to measure the pH of lysosomes. This dye permeates live cells and accumulates in the lysosomes where it shifts fluorescence from blue to yellow depending on the acidity. For quantification of the intra-lysosomal pH, cells plated on a black 96-well plate (1.5 × 10^4^ cells/well) and treated as designated were loaded with 1 μM LysoSensor™ Yellow/Blue DND-160 for 5 min. A pH calibration curve using pH values of 4.0, 4.5, 5.0, 5.5, 6.0, and 7.0 was then generated by incubating cells with standard buffers of known pH containing 10 μM nigericin for 10 min, as described previously [62]. The ionophore nigericin equilibrates pH across cells so that the final ion gradients depend on the experimental conditions [63]. Fluorescence intensity was measured in triplicate using the Varioskan LUX Multimode Microplate Reader (Thermo Fisher Scientific) for light emitted at 440 nm and 540 nm with reference excitation wavelengths of 329 nm and 384 nm, respectively. The lysosomal pH of samples was determined with the help of the calibration curve.

### 4.5. Mitochondrial Membrane Potential

Mitochondrial membrane potential was measured using the tetramethylrhodamine, ethyl ester (TMRE) potentiometric probe according to the manufacturer’s protocol (cat # 87917, Sigma-Aldrich). In brief, ARPE-19 cells seeded at approximately 1 × 10^4^ cells/well on a 96-well plate were grown for 48 h. Cells treated as designated or with the negative control, carbonyl cyanide 4-(trifluoromethoxy) phenylhydrazone (FCCP), for 10 min [64] were incubated with 400 nM TMRE dye at 37 °C for 15 min, followed by repeat rinsing with 1 × PBS/0.2% BSA. Fluorescence at excitation/emission wavelengths of 549/575 nm was quantified using a Clariostar microplate reader (BMG Labtech). Data were normalized with Hoechst stain fluorescence.

### 4.6. Mitochondrial Morphology Using Confocal Microscopy

An assessment of the mitochondrial morphology was performed using the fluorescent mitochondrial marker MitoTracker Green FM (M7514, Thermo Fischer Scientific) which is stable and unaffected by changes in mitochondrial membrane potential [65]. Briefly, cells were seeded onto 35 mm MatTek glass-bottom dishes (1 × 10^6^cells/dish), cultured for 48 h, and treated as indicated, followed by loading with 50 nM MitoTracker Green FM. After a 15-min incubation in the dark, cells were rinsed and visualized in a serum-free medium under an inverted confocal microscope (Eclipse Ti2-E, Nikon Instruments Europe B.V., Amsterdam, The Netherlands) using a 63× magnification with excitation/emission wavelengths of 490/516 nm.

### 4.7. Flow Cytometry with Annexin V-FITC/PI

Cellular apoptosis was assessed using flow cytometry and the Annexin V Apoptosis Detection Kit (Cat # 640914, BioLegend Inc., San Diego, CA, USA) using the manufacturer-recommended propidium iodide (PI) double staining approach. Briefly, trypsinized cells were rinsed twice, and cells were suspended at a density of 1 × 10^6^cells/mL incubated with 5 μL Annexin V-FITC for 10 minutes in the dark at room temperature, followed by incubation with 10 μL PI for 5 minutes and dilution with 400 µL binding buffer for flow cytometry (BD FACSVia Flow Cytometer, BD Biosciences, Franklin Lakes, NJ, USA). Unstained cells in the presence of dyes (FITC−, PI−) are viable; the FITC-stained cells (FITC+, PI−) are undergoing early apoptosis, and double-stained cells (FITC+, PI+) indicate late apoptosis or necrosis [66].

### 4.8. Autophagosome Accumulation Assessment by GFP-LC3 Puncta

Expression of GFP-LC3 in ARPE-19 cells at a cell density of 1.0 × 10^6^ cells was performed by transfection with 2.5 μg pEGFP-LC3 plasmid (Addgene plasmid # 24920) using Lipofectamine 3000 (Invitrogen) for 24 hours in a 35 mm confocal dish. After pharmacological treatment, autophagy flux was assessed by quantifying the GFP-LC3 puncta number per cell using an inverted confocal microscope (Eclipse Ti2-E, Nikon Instruments Europe B.V., Amsterdam, The Netherlands) with a 63× objective. For each treatment condition, the average GFP-LC3 puncta per cell were determined by counting 30 cells.

### 4.9. Protein Carbonyl Assay

Spectrophotometry measurement of the protein carbonyl level was performed using an ELISA kit following the manufacturer’s instructions (Oxiselect™ protein carbonyl, STA-310, Cell Biolabs, San Diego, CA, USA). Briefly, lysate from cells at cell density of 1.5 × 10^6^ cells/well treated as designated in a 6-well plate was incubated with 1% streptomycin sulfate (S9137, Sigma-Aldrich), and protein extract at a concentration of 10 µg/mL was adsorbed onto a 96-well plate for 2 h at 37 °C, and protein carbonyls were then derivatized to DNP hydrazine. Finally, samples were incubated with an anti-DNP antibody followed by HRP conjugated secondary antibody, and the absorbance was measured at 450 nm wavelength using a microplate Reader plate (Ao, Azure Biosystems Inc., Dublin, CA, USA).

### 4.10. Proteasome Activity Assay

Proteasome activity was measured using a fluorogenic 7-amino-4-methyl coumarin (AMC)-tagged substrate kit to detect chymotrypsin-like activity following the manufacturer’s protocol (Cat #: K245, Biovision, San Francisco, CA, USA). Briefly, lysate from treated cells was extracted using 25 mM Tris-HCl buffer and loaded onto a 96-well plate in duplicate for incubation with the fluorescent substrate at 37 °C for 30 min in the presence of MG132 (proteasome inhibitor) or without (as control). Due to the chymotrypsin-like activity of proteasomes, highly fluorescent AMC is released from the AMC-tagged peptide substrate. Fluorescence intensity was then measured at excitation/emission of 350/440 nm using a Clariostar microplate reader (BMG Labtech). Results were normalized to the protein concentration of samples.

### 4.11. Western Blot

A 30 µg denatured protein sample was loaded onto each well of a separating gel for SDS-PAGE electrophoresis (10% SDS-PAGE gels). Electro-transfer of proteins from gel to an Immobilon-FL PVDF membrane (Millipore) took 2 h in prechilled buffer with cold pack using 250 mA. Membrane blocking involved incubation with 5% non-fat milk in Tris-buffered saline containing 0.05% Tween 20 (Bio-Rad Laboratories) for 1 h at room temperature. Primary antibody incubation with anti-LC3 (NB100-2220, Novus Biologicals, Littleton, CO, dilution 1:1000), anti-LAMP2 (sc-18822, Santa Cruz Biotechnology, Dallas, TX, USA, dilution 1:2000), anti-cathepsin D (sc-377299, Santa Cruz Biotechnology, dilution 1:500), anti-TFEB (D2O7D, Cell Signaling Technology, Davers, MA, USA, 1:500), and β-actin (AC-15, Thermo Fisher Scientific, dilution 1:2000) was performed overnight at 4 °C. The washed membrane was incubated with horseradish peroxidase HRP-conjugated secondary antibodies including anti-mouse IgG (H + L, A16066) and anti-rabbit IgG (H + L, A16110; Thermo Fisher Scientific, dilution 1:2000) for 1 h, washed, and developed by incubation with ECL substrate solutions for 5 min. The western blot images were acquired using the Chemidoc MP Imaging System (Bio-Rad, Hercules, CA, USA).

### 4.12. shRNA Knockdown of ATG5

Stable knockdown of *ATG5* in ARPE-19 cells was achieved using lentiviral delivery of short hairpin RNA (shRNA). HEK293T cells seeded in a 10 cm culture dish (3 × 10^6^ cell/dish) were transfected with a lentiviral vector coding a scrambled shRNA plasmid (Addgene plasmid # 1864) or ATG5 shRNA, TRC numbers: TRCN0000151474 (Sigma-Aldrich) using Lipofectamine 2000 (Invitrogen). The transfection lasted for 8 h followed by incubation in a fresh medium for 48 h. Virions were collected and precipitated overnight using PEG before filtering with a 0.45 μm filter. ARPE-19 cells were incubated with virions for 48 h for cell transduction, followed by treatment with puromycin (1.0 μg/mL) for 10 days for the identification of transduced, puromycin-resistant colonies.

### 4.13. RNA Extraction and Quantitative RT-PCR

cDNA was synthesized (High Capacity cDNA Reverse Transcription Kit, Thermo Fisher Scientific) from 1 μg total RNA extracted using Trizol (Invitrogen) as described previously [18]. The reaction mixture used in quantitative RT-PCR contained 2 μL cDNA template, 5 μL LightCycler 480 SYBR Green I Master mix (Roche Diagnostics, Mannheim, Germany), 1 μL nuclease-free water, and 1 μL of gene-specific primers. Primer sequences were as follows: *ATG5* forward: 5′-AAGCTGTTTCGTCCTGTGGC-3′ and *ATG5* reverse: 5′-CCGGGTAGCTCAGATGTTCA-3′; *ATG7* forward: 5′-CGTTGCCCACAGCATCATCTTC-3′ and *ATG7* reverse: 5′-TCCCATGCCTCCTTTCTGGTTC-3′; *β-actin* forward: 5′-CCAACCGCGAGAAGATGA-3′ and *β-actin* reverse: 5′-CCAGAGGCGTACAGGGATAG-3′. The conditions used to run the LightCycler^®^480 Instrument II (Roche Diagnostics) included denaturation at 95 °C for 5 min, followed by 40 cycles at 95 °C for 30 s, 60 °C for 30 s, and 72 °C for 30 s. Fold changes were calculated using the change in the Cycle threshold (*∆∆*CT) method. *β-actin* was used for normalizing the expressions of other genes following the validation of its stability by the coefficient of variation analysis (CV) in ARPE-19 cells under normal and treatment conditions.

### 4.14. Data Analysis

GraphPad Prism (Graphpad Software Inc., San Deigo, CA, USA) was used for analyzing data. All data are presented as the mean ± SD. In determining the difference between treatments, an unpaired *t*-test was used for two treatment groups, and one-way ANOVA followed by Sidak’s/Dunnett’s multiple comparison *post hoc* tests was performed when three or more treatment groups were involved. *p* < 0.05 indicates statistical significance.

## Figures and Tables

**Figure 1 ijms-22-09042-f001:**
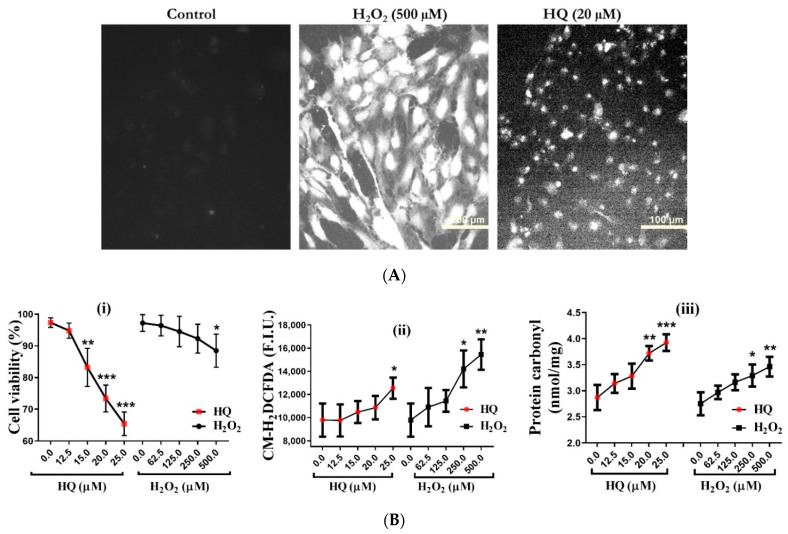
ARPE-19 cells are vulnerable to hydroquinone (HQ)-induced oxidative damage. (**A**) Live-cell confocal microscopy for intracellular reactive oxygen species (ROS) using CM-H2DCFDA dye after incubation of cells with HQ or H_2_O_2_ for 2 h. Basal ROS level in the control is barely visible compared to H_2_O_2_ treatment or HQ upon calibration. (**B**) Quantification of the (i) cell viability by Trypan blue dye exclusion assay, (ii) ROS levels, and (iii) protein carbonyl levels by fluorescent spectrometry using a microplate reader following incubation with HQ or H_2_O_2_ for 2 h. Data represent the mean (+standard deviation, SD) of 3 independent experiments of 3 replicates each. Statistical analysis was performed by one-way ANOVA followed by Dunnett’s multiple comparison tests. * *p* < 0.05, ** *p* < 0.01, *** *p* < 0.001, significant difference relative to the controls. Fluorescence intensity units (F.I.U.).

**Figure 2 ijms-22-09042-f002:**
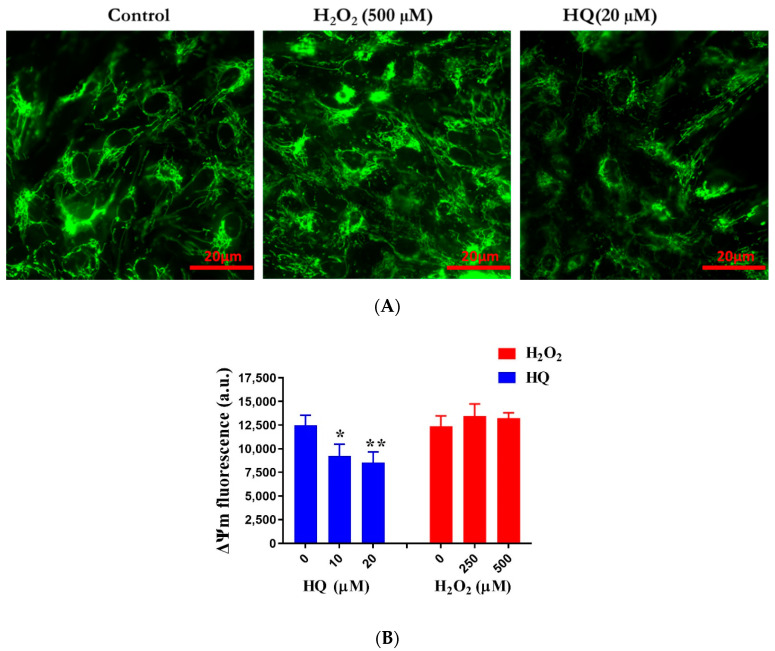
Effects of hydroquinone (HQ) and H_2_O_2_ on mitochondrial morphology and mitochondrial membrane potential. (**A**) Live-cell fluorescence microscopy with the MitoTracker Green FM dye to determine changes in mitochondrial morphology in cells after incubation with HQ or H_2_O_2_ for 2 h. (**B**) Measurement of mitochondrial membrane potential using TMRE dye in cells after treatment with HQ or H_2_O_2_ for 2 h and using a fluorescence microplate reader at excitation/emission of 549 nm/575 nm. Data represent the mean (+standard deviation, SD) of 3 independent experiments of 3 replicates each. Statistical analysis was performed by one-way ANOVA followed by Dunnett’s multiple comparison tests. * *p* < 0.05, ** *p* < 0.01, significant difference relative to the controls. Arbitrary unit (a.u.).

**Figure 3 ijms-22-09042-f003:**
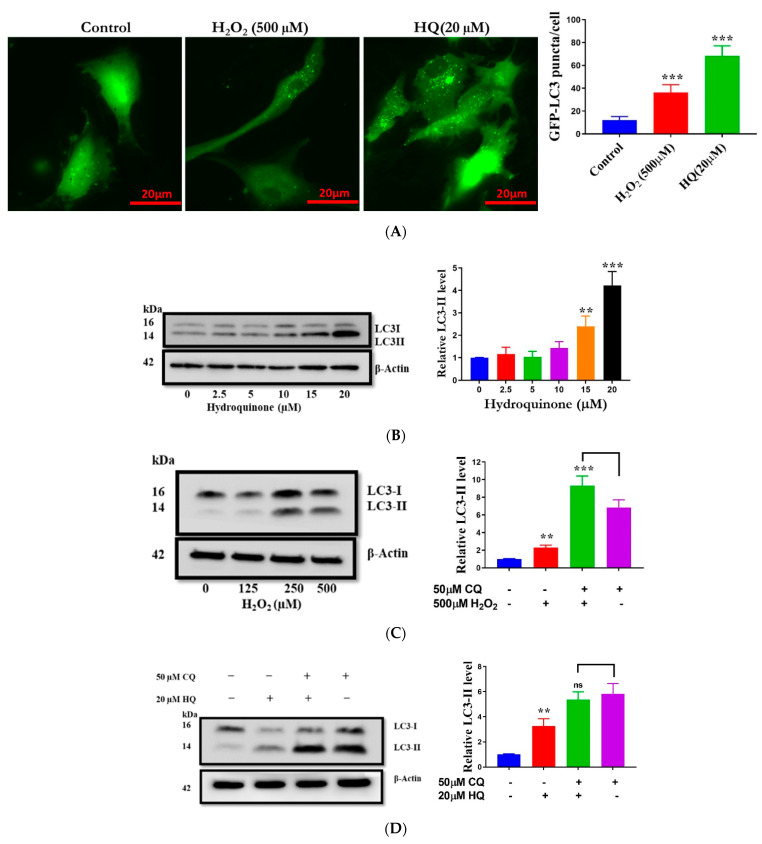
Differential effects of hydroquinone (HQ) and H_2_O_2_ on autophagy flux in ARPE-19 cells. (**A**) Increased GFP-LC3 puncta indicating the accumulation of autophagosomes in cells transfected with the GFP-LC3 plasmid and incubated with HQ or H_2_O_2_ for 2 h. (**B**,**C**) Immunoblot showing a dose-dependent increase in the autophagy marker LC3-II in ARPE-19 cells treated with HQ or H_2_O_2_ for 2 h. (**D**,**E**) Autophagy flux in cells with chloroquine (CQ) pretreatment for 8 h, followed by incubating with HQ or H_2_O_2_ for 2 h. Densitometry quantification of protein levels was normalized with β-actin and expressed as a ratio relative to the control. Data represent the mean + SD of 3 independent experiments of 3 replicates each. Statistical analysis using one-way ANOVA followed by Sidak or Dunnett’s multiple comparison test. ** *p* < 0.01, *** *p* < 0.001 vs control, significant difference relative to the controls.

**Figure 4 ijms-22-09042-f004:**
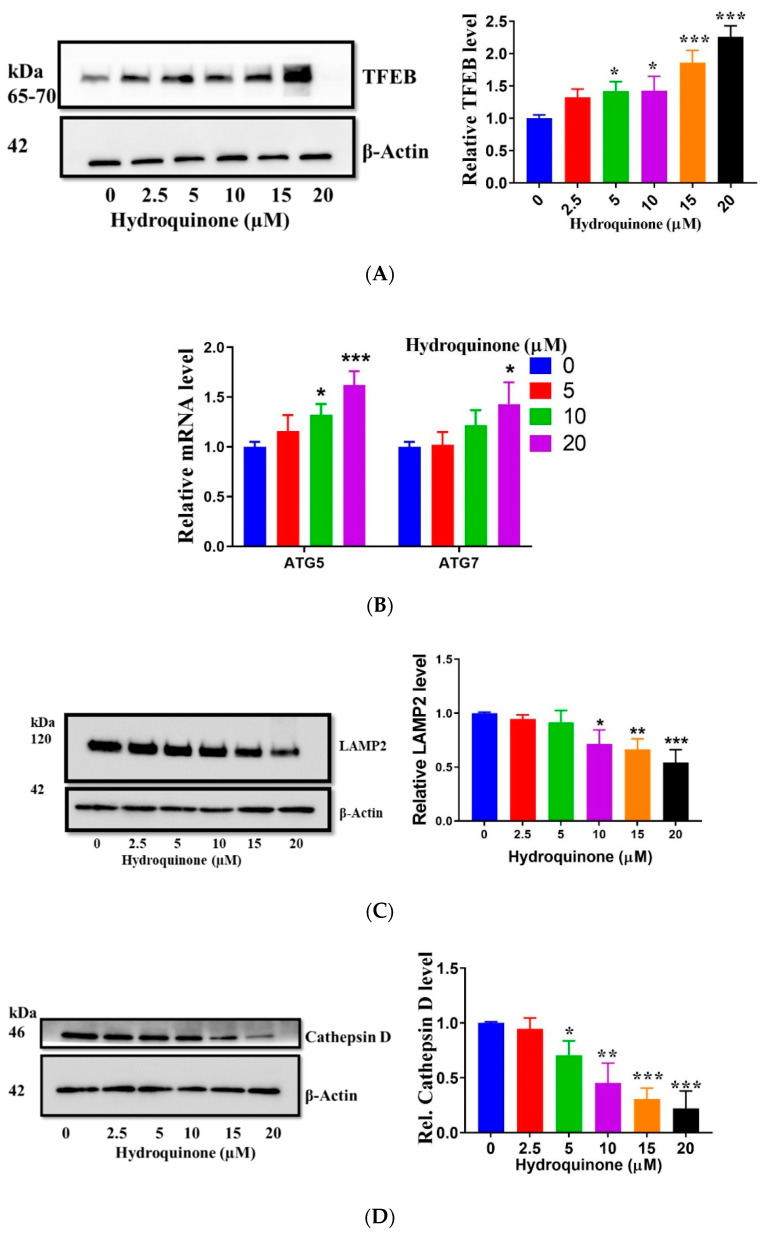
Effect of hydroquinone (HQ) on mRNA and protein markers of the autophagy-lysosomal pathway. (**A**,**B**) Overexpression of TFEB protein and elevated mRNA levels of ATG5 and ATG7 in ARPE-19 incubated with HQ for 2 h. (**C**,**D**) Immunoblots for LAMP2 and cathepsin D levels in whole-cell lysate from cells exposed to HQ for 2 h. Densitometric quantification of protein levels was normalized to β-actin and expressed as a ratio of the control. Data represent the mean + SD of 3 independent experiments of 3 replicates each. Statistical analysis using one-way ANOVA followed by Dunnett’s multiple comparison test. * *p* < 0.05, ** *p* < 0.01, *** *p* < 0.001, significant difference relative to the controls.

**Figure 5 ijms-22-09042-f005:**
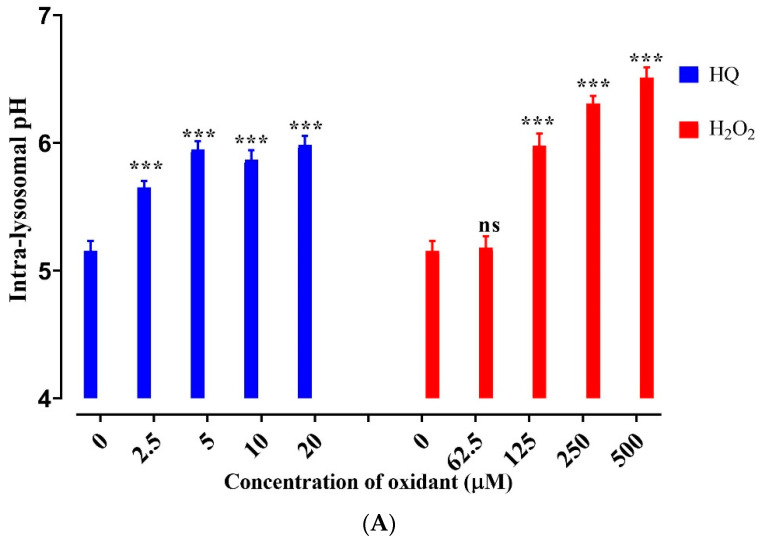
Effect of oxidants on lysosomal pH and proteasome activity. (**A**) Measurement of intra-lysosomal pH using LysoSensor™ Yellow/Blue DND-160 in ARPE-19 cells plated in 96-well plates and incubated with HQ or H_2_O_2_ for 2 h. (**B**,**C**) Proteasome activity using a fluorogenic substrate for detection of chymotrypsin-like activity in cells treated with HQ or H_2_O_2_ for 2 h. Data represent the mean + SD of 3 independent experiments of 3 replicates each. Statistical analysis using one-way ANOVA followed by Dunnett’s multiple comparison test. ** *p* < 0.01, *** *p* < 0.001, significant difference relative to the controls.

**Figure 6 ijms-22-09042-f006:**
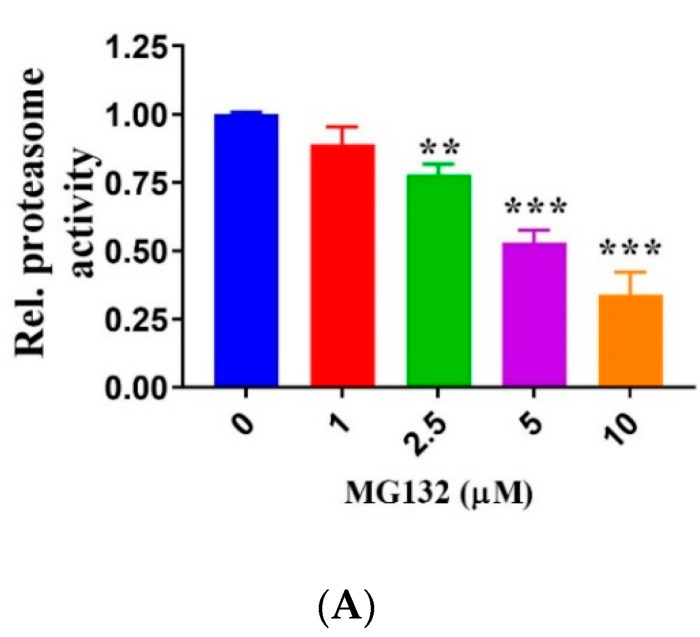
Effect of MG132 on proteasome inhibition and apoptosis in ARPE-19 cells. (**A**) Proteasome activity declines dose-dependently in cells treated with MG132 for 5 h. (**B**) Annexin V-FITC/PI staining to assess cell death after treatment with MG132 or vehicle (control) for 5 h. Data represent the mean (+ SD) of 3 independent experiments of 3 replicates each. Statistical analysis using one-way ANOVA followed by Dunnett’s multiple comparison test. ** *p* < 0.01, *** *p* < 0.001, significant difference relative to the controls.

**Figure 7 ijms-22-09042-f007:**
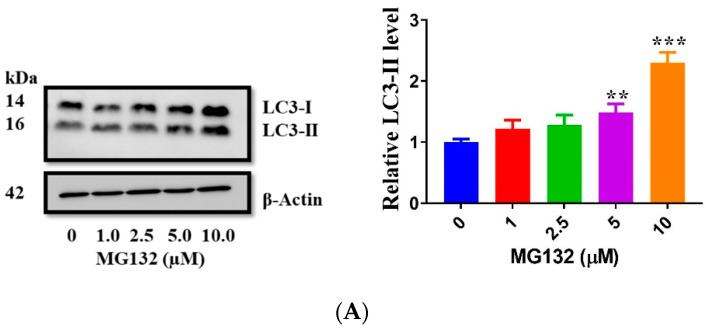
MG132 upregulates autophagy and LAMP2 expression in human RPE cells. (**A**) LC3-II levels increased in ARPE-19 cells after treatment with MG132 for 5 h. (**B**) LC3-II levels in cells incubated with chloroquine for 8 h followed by incubating with MG132 for 5 h. (**C**) LAMP2 levels increased in cells after 5 h incubation with MG132. Protein levels were normalized with β-actin and expressed as a ratio of the control. Data represent the mean (+SD) of 3 independent experiments of 3 replicates each. Statistical analysis using one-way ANOVA followed by Dunnett’s or Sidak’s multiple comparisons test. * *p* < 0.05, ** *p* < 0.01, *** *p* < 0.001, significant difference relative to the controls.

**Figure 8 ijms-22-09042-f008:**
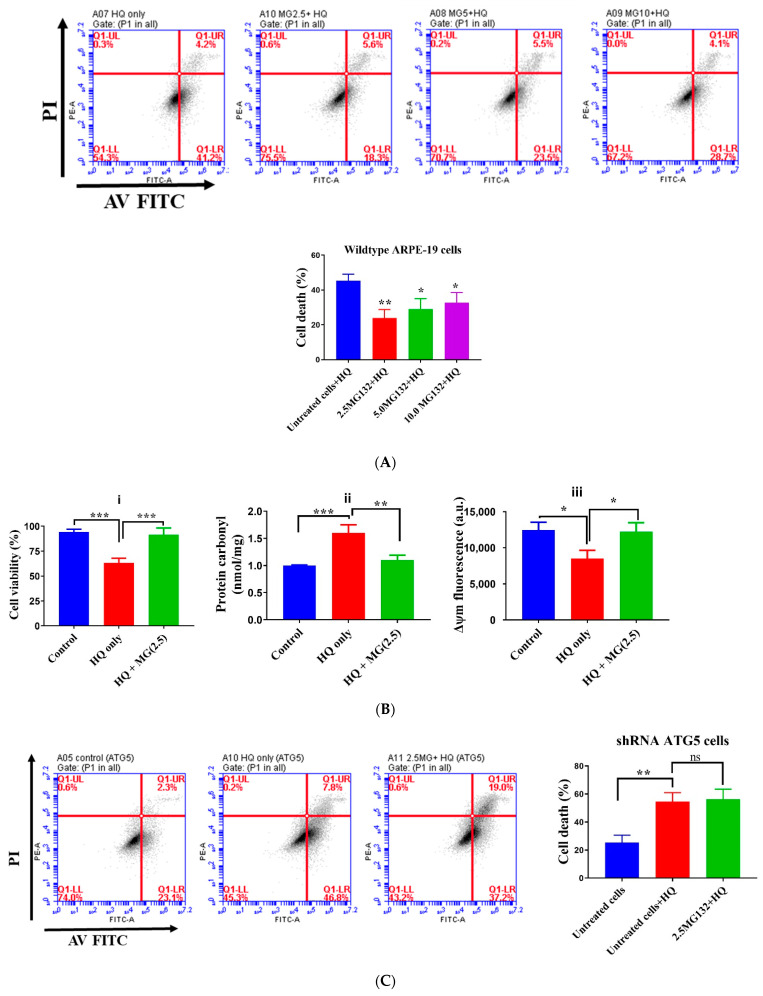
MG132 protects against hydroquinone-induced oxidative damage and apoptosis in autophagy competent cells. (**A**) Flow cytometry using annexin V-FITC/PI staining in cells treated with MG132 or vehicle (ethanol) for 3 h before incubation with 25 µM HQ for 2 h. (**B**) Measures of cell viability, protein carbonyl level, and mitochondrial membrane potential in cells pretreated with 2.5 µM MG 132 or vehicle for 3 h before incubation with 25 µM HQ for 2 h. (**C**) Autophagy inhibition following transfection of shRNA against ATG5 caused loss of MG132-mediated protection against HQ-induced apoptotic damage in cells. Data represent the mean (+SD) of 3 independent experiments of 3 replicates each. Statistical analysis using one-way ANOVA followed by Sidak or Dunnett’s multiple comparison test. * *p* < 0.05, ** *p* < 0.01, *** *p* < 0.001, significant difference relative to the control.

## Data Availability

All data supporting the results of the study have been presented as figures.

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
