# Peer review of "Targeting Lysosomes to Reverse Hydroquinone-Induced Autophagy Defects and Oxidative Damage in Human Retinal Pigment Epithelial Cells"

_ijms, 2021, doi:10.3390/ijms22169042_

Round 1
Reviewer 1 Report
Ref: ijms-1323391
Title: Targeting lysosomes to reverse hydroquinone-induced autophagy defects and oxidative damage in human retinal pigment epithelial cells
Recommendation: Minor revision
The experiments are planned and designed correctly, researchers have used wide variety of methods to support the results of the investigations, Results of this valuable work are presented correctly. The Materials and Methods are described very carefully. Nevertheless, I have some questions and some request to be fulfill before publication.
- Please provide the full western blots.
- The special characters such as µ, β etc. look bad on the graphs. Please, correct it.
- Please provide the “n” number to each experiment as well as the number of replicates.
- Please provide the density of the cell cultures to paragraph 4.8. and 4.9.
- How the Authors chose the b-actin as a reference gene in qPCR method? What kind of evaluation and screening they use?
Reviewer 2 Report
This paper presents the finding that lysosomal dysfunction and autophagy deficits represent the mechanisms underlying HQ-induced oxidative damage in human RPE. Moreover, the authors suggest that lysosomal stabilization, via proteasome inhibition could represent a therapeutical strategy against RPE oxidative damage and AMD.
The experimental design of the manuscript is well-conducted, but the manuscript contains many inaccuracies and requires a revision before it can be considered for publication.
It is necessary that the authors improve the introduction section, by adding more information regarding the pathophysiological underlying mechanism AMD.
In figure 4A, the expression of TFEB protein does not seem significantly higher when cells are treated with 20 uM of HQ as compared to 10 or 15 uM of HQ. Please check and provide another representative western blot image.
Figure 4D, the quality of wb image is too poor and not suitable for publication. Please provide high quality images of the blot.
Author Response
Title: Targeting lysosomes to reverse hydroquinone-induced autophagy defects and oxidative damage in human retinal pigment epithelial cells
Comments and Suggestions for Authors
Comment: This paper presents the finding that lysosomal dysfunction and autophagy deficits represent the mechanisms underlying HQ-induced oxidative damage in human RPE. Moreover, the authors suggest that lysosomal stabilization, via proteasome inhibition could represent a therapeutical strategy against RPE oxidative damage and AMD.
The experimental design of the manuscript is well-conducted, but the manuscript contains many inaccuracies and requires a revision before it can be considered for publication.
Response: Thanks for the comments. Please, find below the revisions done in the revised manuscript.
Comment: It is necessary that the authors improve the introduction section, by adding more information regarding the pathophysiological underlying mechanism AMD.
Response: AMD pathophysiology has been included (paragraph 4, lines 7-11). Thanks
Comment: In figure 4A, the expression of TFEB protein does not seem significantly higher when cells are treated with 20 uM of HQ as compared to 10 or 15 uM of HQ. Please check and provide another representative western blot image.
Response: We have replaced it with a more representative western blot micrograph. Thanks.
Comment: Figure 4D, the quality of wb image is too poor and not suitable for publication. Please provide high quality images of the blot.
Response: Replaced with a better quality western blot micrograph, as suggested.
Round 2
Reviewer 2 Report
The authors responded well to all requests.
Author Response
Thanks for the favorable assessment of this manuscript.